# Earthworms contribute significantly to global food production

Steven J. Fonte [1] ✉, Marian Hsieh [2] & Nathaniel D. Mueller[1,2]

Earthworms are critical soil ecosystem engineers that support plant growth in numerous ways; however, their contribution to global agricultural production has not been quantified. We estimate the impacts of earthworms on global production of key crops by analyzing maps of earthworm abundance, soil properties, and crop yields together with earthworm-yield responses from the literature. Our findings indicate that earthworms contribute to roughly 6.5% of global grain (maize, rice, wheat, barley) production and 2.3% of legume production, equivalent to over 140 million metric tons annually. The earthworm contribution is especially notable in the global South, where earthworms contribute 10% of total grain production in Sub-Saharan Africa and 8% in Latin America and the Caribbean. Our findings suggest that earthworms are important drivers of global food production and that investment in agroecological policies and practices to support earthworm populations and overall soil biodiversity could contribute greatly to sustainable agricultural goals.

Agricultural intensification (e.g., via improved crop varieties, agrochemical inputs, mechanized tillage) is largely credited for feeding a growing population during the last century, however, these changes have come at a significant environmental cost in terms of biodiversity loss, water and air pollution, climate change, and multiple other side effects[1]. These problems are only expected to intensify as global food demand continues to rise, thus emphasizing the need for more agroecological management approaches to produce our food[2].

The sustainable management of soils and overall soil health represents a key element in agroecological intensification efforts, as soil biological communities offer great potential to support food production and a range of other ecosystem services[3]. Earthworms, in particular, are important ecosystem engineers that influence plant growth via impacts on soil structure, water capture, organic matter cycling, and nutrient availability[4,5]. Earthworms have also been shown to facilitate the production of plant growth-promoting hormones and trigger effective crop immune responses to common soil pathogens[6]. Despite their widespread recognition as indicators and builders of healthy soils, the potential contribution of earthworms and other beneficial soil organisms to global agricultural production remains poorly understood[3,7], yet such knowledge is fundamental for the innovation of new agroecological practices and policies.

We estimate earthworm impacts on the productivity of major cereal and legume crops using mean effect sizes from a meta-analysis that reported varying earthworm-yield responses for different crop types, soil properties (texture, pH), N fertilizer inputs, as well as at different levels of earthworm abundance[8]. These values are combined with a recent map on the distribution and abundance of earthworms[9] together with global soil, management, and crop data layers to estimate the potential contribution of earthworms to agricultural production at a global scale.

## Results and discussion

Our findings suggest that earthworms contribute roughly 5.4% of global production for the major cereal and legume crops considered in this study. When looking just at common cereal crops (rice, maize, wheat, barley), the contribution of earthworms is estimated to be 6.45% of global production, or roughly 128 million metric tons of grain. Earthworm impacts on legumes (e.g., soybean, dry beans, peas, garbanzos, lentils, alfalfa, clover) were lower, contributing just 2.3% of the global total, or 16 million metric tons. The large difference between cereals and legumes is not surprising, and is largely due to the more pronounced effects of earthworms on the growth of cereals reported by van Groeningen et al.[8], which likely results from the capacity of legumes to fix

[1]Department of Soil and Crop Sciences, Colorado State University, Fort Collins, CO, USA. [2]Department of Ecosystem Science and Sustainability, Colorado State University, Fort Collins, CO, USA. ✉e-mail: steven.fonte@colostate.edu

their own N, and thus not benefiting to the same degree as cereals from earthworm-facilitated mineralization of organic N in the soil.

## Regional impacts of earthworms

When considering earthworm impacts across different regions, we observed the greatest relative effect in Sub-Saharan Africa, where earthworms are estimated to contribute roughly 10% of total cereal production and 3.2% of legume production (Figs. 1 and 2a). This was followed by Latin America and the Caribbean, where roughly 8% of cereal grain and 3.1% of legume production can be attributed to earthworms. The higher impact of earthworms on yields in the global South appears to be associated with soils generally having lower pH and higher clay content, as well as lower fertilizer inputs, all factors shown to enhance the relative benefit of earthworms to plant growth[8]. We note that estimated earthworm impacts in Europe and Eastern/South-Eastern Asia were also relatively high, with 7.4% of cereal grain production attributable to earthworm activity. The higher-than-average effect in these regions is related to higher predicted earthworm abundance[9], especially in the case of Europe, as well as lower than average soil pH values, particularly in South-Eastern Asia. Meanwhile, less pronounced earthworm impacts in other regions are likely associated with low estimated earthworm abundance, higher rates of inorganic fertilizer use and/or soil properties that lessen the observed benefit of earthworm additions.

Our estimates for total production increases tell a somewhat different story (Fig. 2b), due to the vastly different levels of production in different regions around the globe. Earthworm contributions are estimated to be highest in Eastern/South-Eastern Asia and Europe, with over 40 million metric tons of cereal grain production attributable to earthworm activity in each of these regions. This result comes from the above-average earthworm contribution to yield, together with higher overall productivity and expansive cropping areas in these regions. At the same time, low overall agricultural production in both Sub-Saharan Africa and the Latin America and Caribbean region translates to considerably lower earthworm-associated increases in total production in these regions (4.0 and 10.7 million metric tons of cereal grain; respectively). Despite the smaller absolute contribution of earthworms to agricultural production in the global South, even modest increases in these regions are likely to be much more important for addressing issues of hunger and malnutrition, and thus should not be discounted.

## Recognizing sources of uncertainty

Our results are encouraging and suggest significant potential to enhance agricultural productivity via improved management of soil biological communities. However, we recognize that these estimates include several key potential sources of error that need to be acknowledged. First, we note that most of the studies used in the meta-analysis[8] relied on simple mesocosms, typically with earthworm

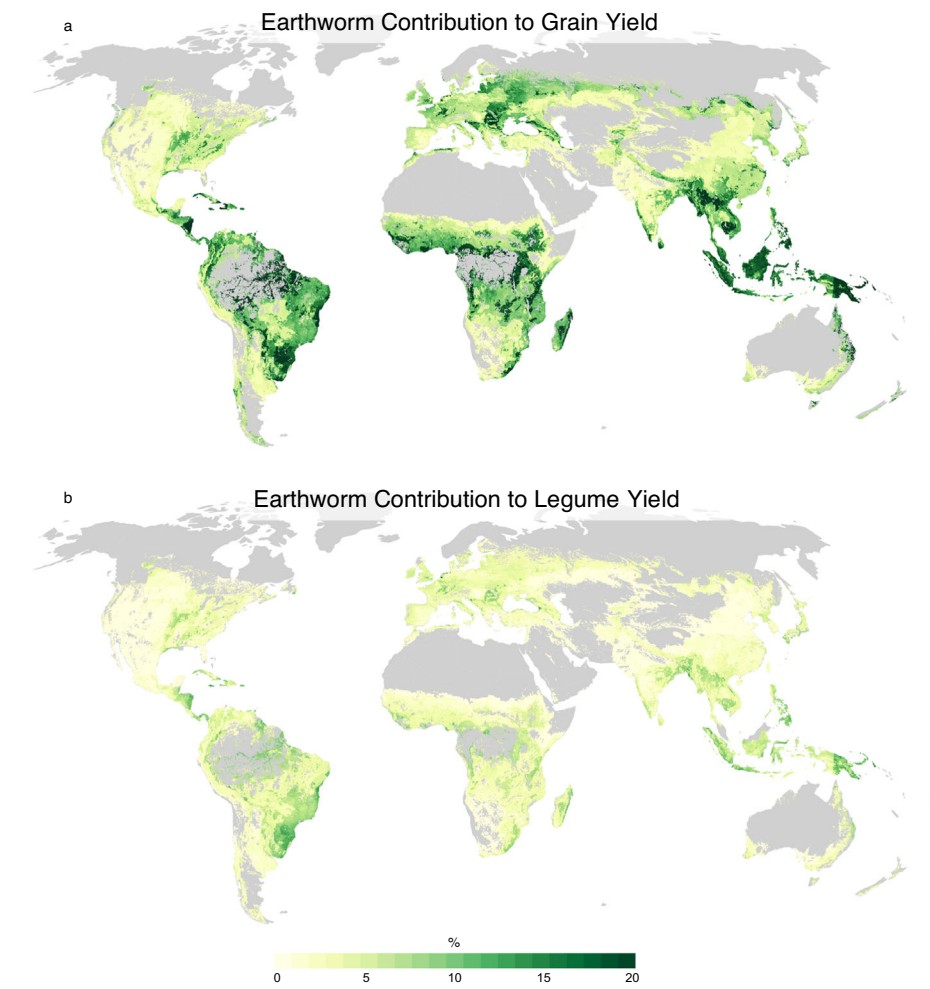

**Fig. 1 | Earthworm contributions to global yields.** Relative contribution of earthworms to yield (% of total) of **a** cereal grains (i.e., grass species), and **b** legume yields. Grains considered include wheat, rice, maize, and barley. Legumes considered include grain legumes (soybean, dry beans, broad beans, cowpeas, peas, pigeon peas, chickpeas, lentils, lupines, and other pulses) as well as forage species, alfalfa and clover (see Figs. S1 and S2). Darker shades of green indicate stronger estimated earthworm impacts.

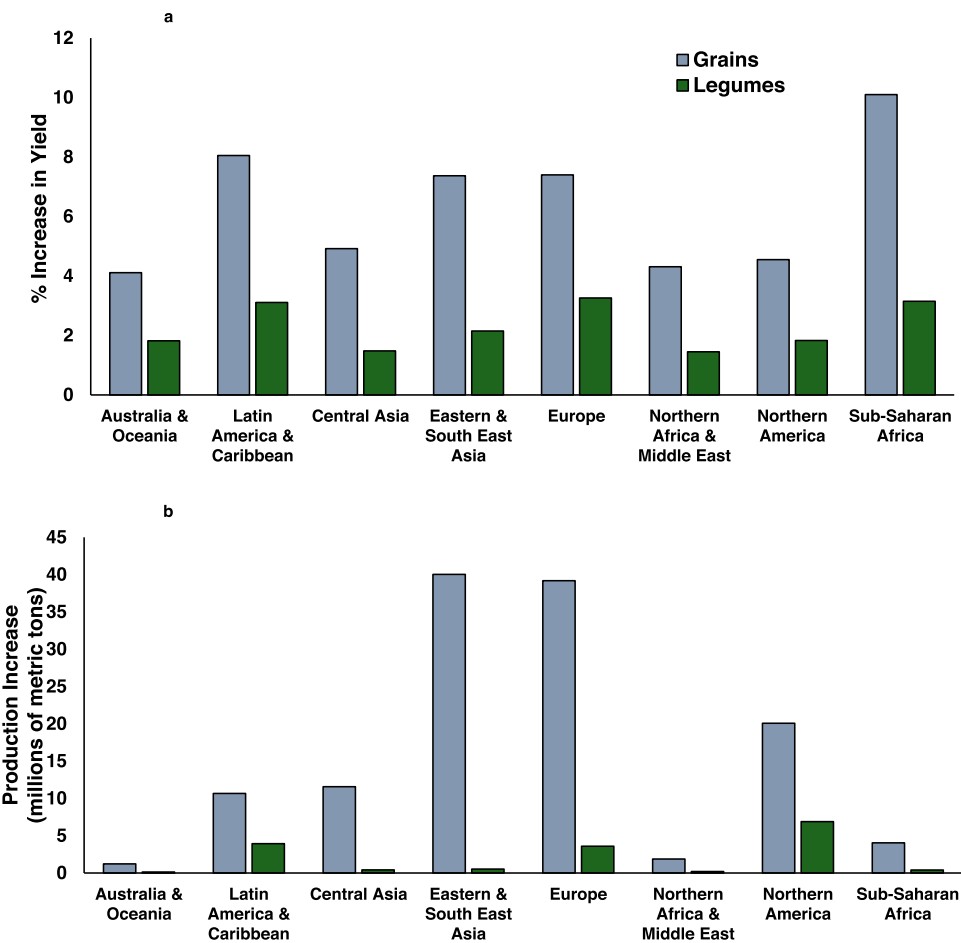

**Fig. 2 | Regional contribution of earthworms to crop production.** Contribution of earthworms to grain and legume production across eight global agricultural regions in: **a** relative yield increase (% of total), and **b** absolute production increase (millions of metric tons).

additions (and/or exclusion), to estimate the effect of earthworms on plant growth. While this offers a reasonable approach to examine earthworm benefits to plants, these studies often tend to test high, sometimes unrealistic, densities of earthworms and may thus overestimate the potential earthworm effect on plant growth. We attempted to correct for this in our analysis by taking abundance into account when calculating the relative earthworm effect, such that soils with lower densities of earthworms had much lower estimated productivity increases. At the same time, we note that the short-term nature of these experiments means that they do not capture the full array of benefits that earthworms and other soil macrofauna can have on multiple soil functions (e.g., erosion control, water dynamics), which can contribute to improved crop growth in the long-term, under more realistic conditions[7,10,11]. In addition, we point out that our analysis assumed simple additive effects for the plant and environmental factors that influenced the earthworm benefit, as we were not able to parse out potential interactions between these drivers.

While the map of earthworm abundance used here offers a valuable first approximation of earthworm populations around the globe[9], there was a strong sampling bias towards the global North, with the vast majority of data points coming from Europe and eastern North America. We also note that abundance estimates from much of the global South are quite low, especially in poorly represented areas such as Sub-Saharan Africa. This indicates two possibilities—first, we could interpret this to mean that the regions with low earthworm abundance have great potential to benefit from the adoption of agricultural practices that promote earthworm populations (e.g., no-till agriculture, increased organic inputs/return). Alternatively, the global

earthworm map used here may underestimate both the diversity and abundance of earthworms in these regions, especially in the tropics[12], and earthworm contributions across the global South could already be much higher than estimated here. In this same vein, we acknowledge that all global maps, including those used in our analyses, are subject to varying degrees of uncertainty[13], and this becomes further compounded when data layers of differing resolutions are combined to generate new maps.

## Conclusions

This is the first effort to our knowledge that attempts to quantify the contribution of a beneficial soil organism to global agricultural production. While the effect of earthworms is notable, we suspect that other soil biota may be equally as important and that further study is needed. Also, to be clear, we do not advocate for the widespread inoculation of earthworms to regions where they are not currently present, as this can have highly undesirable ecological consequences for adjacent natural areas[14,15]. Instead, we suggest investment in continued research and promotion of agroecological management practices that enhance entire soil biological communities, including earthworms, so as to support a whole range of ecosystem services that contribute to the long-term sustainability and resilience of agriculture.

## Methods
### Data layers
Global crop yields and harvest areas (at 5 arcminute resolution) were taken from ref. 16. Crops selected were based off of the most common cereal grains reported by van Groeningen et al.[8] (wheat, rice, maize,

**Table 1 | Factors and categories considered in our analysis**

| Factors considered | Categories | Sample size[a] | EW effect[a] | EW effect (% change)[a] | Coefficients |
|---|---|---|---|---|---|
| Crop type | Cereal grains | 106 | + | 31.41 | 1.271 |
|  | Grasses[b] | 154 | + | 24.33 | N/A[b] |
|  | Legumes | 42 | 0 | 9.19 | 0.372 |
|  | Total: | 302 |  | 24.71[c] |  |
| Soil pH | Low ( < 5.6) | 93 | + | 33.43 | 1.235 |
|  | Medium (5.6–7.0) | 76 | + | 33.64 | 1.242 |
|  | High ( > 7.0) | 81 | + | 13.63 | 0.503 |
|  | Total: | 250 |  | 27.08[c] |  |
| Soil texture | Sandy ( > 70% sand) | 16 | 0 | 9.64 | 0.352 |
|  | Loamy ( ≤ 70 % sand & ≤ 35% clay) | 110 | + | 20.74 | 0.758 |
|  | Clayey ( > 35% clay) | 52 | + | 46.79 | 1.711 |
|  | Total: | 178 |  | 27.35[c] |  |
| N application rate | Low ( ≤ 30 kg N ha$^{-1}$ yr$^{-1}$) | 183 | + | 19.00 | 1.071 |
|  | High ( > 30 kg N ha$^{-1}$ yr$^{-1}$) | 25 | 0 | 8.51 | 0.480 |
|  | Total: | 208 |  | 17.74[c] |  |
| Earthworm abundance | Coefficient = 0.1032 × abundance (ind. m$^{-2}$) $^{0.409}$ |  |  |  |  |

[a]Based on effect sizes and confidence intervals (CI) reported in the supplementary information of ref. 8, using estimates based on a parametric weighting function using the inverse of the pooled variance.

[b]Grasses were not included in the analyses due to the lack of an appropriate geographic crop data layer for this group.

[c]Weighted averages of the earthworm effect based on available sample size and effect sizes reported above for each category within a particular factor.

Coefficients are based on sample sizes and earthworm (EW) effect sizes reported by van Groeningen et al.[8], and used to generate a weighted coefficient for each category. For earthworm abundance, the coefficient was developed based on a nonlinear function representing the best fit for earthworm effects reported in the meta-analysis.

and barley) as well as all leguminous annual crops in ref. 16 database, including both grain and forage legumes (see Supplementary Figs. S1 and S2), many of which were included in the meta-analysis[8]. Non-cereal grasses, which were included in the earthworm meta-analysis, were not considered here due to ambiguity of which data layers these should correspond to in the global crop database. Data layers on soil pH and textural classes were taken from SoilGrids[17] (downloaded using Google Earth Engine), using weighted averages of the top five soil layers (0–100 cm in depth). Data for N application rates for each crop were adapted from ref. 18 to differentiate areas receiving low fertilizer N rates ( ≤ 30 kg N ha$^{-1}$ yr$^{-1}$) from those receiving higher rates. Global earthworm distribution was taken from ref. 9 and upscaled by averaging, excluding missing grid cells, to 5 arcminute resolution. The spatial resolution of the earthworm density layer was altered using Matlab R2020a. Analysis was performed with R (Version 4.1.3) in RStudio (Version 2022.12.0 + 353).

**Estimating earthworm impacts**

Our analysis relies on a meta-analysis examining the effect of earthworms on plant productivity[8]. More specifically, the meta-analysis considered studies where earthworms were either added or excluded from greenhouse or field mesocosms and crop responses evaluated. From this meta-analysis, we considered the average effect of earthworms on aboveground plant biomass production (a 23.3% increase) across the 58 studies (462 data points). We then selected the most important drivers of the earthworm effect by examining the effect sizes reported by van Groeningen et al.[8] and considering plant and environmental factors where the categories showed clearly differing earthworm effects (e.g., no-effect vs. positive effect, as reported for high vs. low N application) or had non-overlapping confidence intervals for at least two of the categories (e.g., pH >7 vs. pH ≤7). These variables were then cross-referenced against available global data layers (Supplementary Fig. S3), such that we ignored potential drivers for which there is no available data set (e.g., crop residue application rate). The factors thus selected included: crop type, soil pH, soil texture, nitrogen (N) application rate, and earthworm abundance (individuals m$^{-2}$). For each of these factors, we calculated a coefficient to represent the relative increase in plant production associated with differing levels of each factor (see Table 1). Coefficients were based on the effect sizes reported by van Groeningen et al.[8] (Table 1) and weighted by the number of observations in each category, since this varied depending on the variable in question and availability of data from the original studies. Coefficients for each factor were then used as a multiplier to scale up or down from the average increase of 23.3%. We then generated the overall earthworm effect, $E_{i,k}$, for each grid cell, $i$, and crop, $k$, using the following Eq. (1):

$$E_{i,k} = c_{i,k} * p_i * t_i * n_{i,k} * a_i * 23.3\% \qquad (1)$$

where $c$, $p$, $t$, and $n$ are the coefficients associated with the categories of crop type, soil pH, soil texture, and crop-specific fertilizer N rate, respectively. Earthworm abundance, $a$, was calculated as a continuous, nonlinear (power) function that provided a best fit for the available data (Table 1). We use this approach for earthworms rather than categories of abundance (as reported in the meta-analysis) since earthworm abundance for most cells in the global earthworm map was below the threshold for the 'low' earthworm density category (100 individuals m$^{-2}$) reported by van Groeningen et al.[8]. We consider our approach somewhat conservative since it allows cells with just a few individuals per m$^2$ to have a near zero effect of earthworms.

We calculated maps of $E$ separately for each crop, masked to each crop's production area. To calculate average earthworm effects across crops (Fig. 1), we used crop harvested area to weight at the grid cell level. To estimate the effects of earthworms on absolute crop yields, $E$ was applied to individual crop yield layers to calculate the difference between crop yields with and without earthworm impact. Absolute crop production attributable to earthworms was then calculated by multiplying these two yield maps by the appropriate crop area map and summing the difference in production across grid cells. Grid cells for which there were no earthworm data reported were omitted from the analysis. Finally, the United Nations Sustainable Development Goals regional groupings (with Australia/New Zealand and Oceania combined) were used to calculate impacts on a regional level.

**Reporting summary**
Further information on research design is available in the Nature Portfolio Reporting Summary linked to this article.

**Data availability**
The data generated in this study are available on Zenodo (https://doi.org/10.5281/zenodo.8235224). Global gridded crop harvested area and yield data are from ref. 16, available at http://www.earthstat.org/harvested-area-yield-175-crops/. Global, crop-specific fertilizer data are from ref. 18, available at https://zenodo.org/record/5260732#.YXBFh9nMKDU. Global soil textural classes and pH are from Soil-Grids, available on Google Earth Engine at https://code.earthengine.google.com/32bc92667cf3a553bad8c6cfa4745d55?asset=projects%2Fsoilgrids-isric%2Fphh2o_mean. Global earthworm density data are from ref. 9; a processed version is available in this study's Zenodo repository.

**Code availability**
The scripts used in this study are openly available on Zenodo (https://doi.org/10.5281/zenodo.8235224).

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

## Acknowledgements
We thank Drs. Jan Willem van Groenigen and Ingrid Lubbers for sharing original data collected for their earthworm meta-analysis. We also thank Dr. Helen Phillips for providing updated maps of global earthworm abundance and diversity. Funding was provided to Marian Hsieh from the National Science Foundation Graduate Research Fellowship Program (DGE-006784). This research did not receive any specific grant from funding agencies in the public, commercial, or not-for-profit sectors.

## Author contributions
S.J.F. and N.D.M. conceived the study and the general approach. M.H. and N.D.M. prepared the data and conducted the analyses. M.H. developed visualizations with support from N.D.M. and S.F. S.F. wrote most of the manuscript, with support from N.D.M. and M.H.

## Competing interests
The authors declare no competing interests.
