## [Peer Review File · Nature Communications]

Reviewers' Comments:

Reviewer #1:

Remarks to the Author:

In this manuscript, Fonte et al. use previously published data (a meta-analysis and global data layers) to understand how earthworms contribute to the yield of cereals and legume across the world. They find that on average earthworms have caused yields to be 5.4% higher than if no earthworms were present. The amount varied depending on the crop type, but also global region – due to differences in soil properties and the prevalence of fertilisers. Soil biodiversity, including earthworms, are often cited as having a large impact on food production, and it great to see a synthesis analysis that delves further into this question, and attempts to identify how much impact they have as well as which regions of the world are benefitting the most from a healthy earthworm community. Although articles are being published looking at the contribution of soil biodiversity to ecosystem services (such as food production), these are often focused on microbial diversity, and so it is good to see this analysis focused on the an important member of the soil fauna community.

Overall I think more detail needs to be provided in the methods.

- It is not clear to me where the coefficients have come from in the van Groenigen meta-analysis. The data is not provided, and the code has not information on how those coefficients are derived (and therefore the weighting from the original meta-analysis). If these have been taken from the figures in the main analysis, this should be stated (potentially also including a table of the original coefficients, and the weight values applied by Fonte et al).
- In the underlying meta-analysis, some of the coefficients were not significantly different from zero (e.g., sandy soil did not significantly impact aboveground biomass.). Were non-significant coefficients still used in this synthesis analysis? Assuming so, is this appropriate? My assumption would be not.
- Line 142 "We then selected the most important drivers of the earthworm effect" – it is not clear how this was decided/done. The underlying meta-analysis provides many other coefficients that could have been used in this synthesis analysis (e.g., changes in climate, which are often found to be important). This should be expanded upon in terms of why those predictors were chosen and other not, and what implications this may have.
- Finally, how was the non-linear function for earthworm abundance calculated? In the underlying meta-analysis, there are three coefficients for earthworm abundance, so how robust is a function that has been created from only three data points?

As far as I can tell, the underlying meta-analysis tested each subgroup with a single model (e.g., one model on the effect of soil pH on yield, and another model on the effect of earthworm abundance on yield, etc. etc.). Fonte et al., have assumed that these coefficients (crop type, soil pH, soil texture, and crop-specific fertilizer N rates) can be multiplied together (i.e., a multiplicative effect, line 147). However, one could equally assume that these effects do not significantly interact with each other, and instead could also be causing an additive effect. Unfortunately, the van Groenigen meta-analysis can not be used to determine whether the equation should be multiplicative or additive, so I would suggest that the possibility is tested, and see how much the results change as a result of this assumption.

Minor point:

Line 76-77: "...due *to* the vastly..."

Reviewer #2:

Remarks to the Author:

This study is interesting as it shows clearly that earthworms contribute to food production at the global scale. They showed with a metanalysis that in the whole world the Impact of earthworm on global agriculture is 6.5 % for global grain 2.3 % of legume yield.

It is true that up to now, there is not serious measurements of the earthworm potential contribution to global agricultural production. Most of the information comes from small trials and we don't have the idea of their impact at large scale.

The results of the analysis are interesting, we see that they impact cereals and legumes

differently. As well their impact is more important in the global south although they have impact globally. They suggest that if there is an improved management of soil biological communities, the impact of earthworm will be larger. I Like that they mention that it can be some biases in their analysis which can give some over or under estimations.

The methods used are well explained although I am not an expert on this kind of analysis.

The manuscript is well written and clear. I think this communication is important as it is one of the first work that gives the idea of the impact of earthworms and soil biodiversity can have in food production at a global scale, which is a proof of the importance of agro -and soil biodiversity for decision makers.

So, I recommend the publication of this communication without any corrections,

I hope that over time more data will be available to draw more robust and secure conclusions as their data comes from 54 studies (462) points.

Kind regards

Responses to Referee Comments

Reviewer #1 (Remarks to the Author):

In this manuscript, Fonte et al. use previously published data (a meta-analysis and global data layers) to understand how earthworms contribute to the yield of cereals and legume across the world. They find that on average earthworms have caused yields to be 5.4% higher than if no earthworms were present. The amount varied depending on the crop type, but also global region – due to differences in soil properties and the prevalence of fertilisers. Soil biodiversity, including earthworms, are often cited as having a large impact on food production, and it great to see a synthesis analysis that delves further into this question, and attempts to identify how much impact they have as well as which regions of the world are benefitting the most from a healthy earthworm community. Although articles are being published looking at the contribution of soil biodiversity to ecosystem services (such as food production), these are often focused on microbial diversity, and so it is good to see this analysis focused on the an important member of the soil fauna community.

RESPONSE: We thank the reviewer for their thorough review and encouraging remarks.

Overall I think more detail needs to be provided in the methods.

- It is not clear to me where the coefficients have come from in the van Groenigen meta-analysis. The data is not provided, and the code has not information on how those coefficients are derived (and therefore the weighting from the original meta-analysis). If these have been taken from the figures in the main analysis, this should be stated (potentially also including a table of the original coefficients, and the weight values applied by Fonte et al).

RESPONSE: As the reviewer indicates, the coefficients used in our analysis are largely based on the main figures (effect sizes) presented by van Groenigen et al. (2014) and especially, the supplementary information from their paper, where this data is reported in greater detail. We now explain this better in the methods section by adding the following:

“Coefficients were based on the effect sizes reported by van Groenigen et al.⁸ (Table 1) and weighted by the number of observations in each category, since sample size varied depending on the factor in question and availability of data from the original studies.”

To further address the reviewer’s concern, we now include the relevant sample sizes, overall earthworm effects, and effect size associated with each category (as reported in the supplementary information van Groenigen et al. (2014)), to Table 1 in our manuscript.

- In the underlying meta-analysis, some of the coefficients were not significantly different from zero (e.g., sandy soil did not significantly impact aboveground biomass.). Were non-significant coefficients still used in this synthesis analysis? Assuming so, is this appropriate? My assumption would be not.

RESPONSE: Yes, we recognize that some of the categories/coefficients did not show a significant influence (as is now indicated with zeros in the 'earthworm effect' column of Table 1). However, we felt it would more accurately reflect the data to include coefficients based on the mean values reported by van Groenigen et al. (2014) rather than to simply say there was no earthworm effect. We note that these coefficients are generally low (< 0.5) so should not have an outsized influence on our global estimates of the earthworm effect. Additionally, we note that including a zero for any coefficient in our earthworm effect equation, would essentially nullify the estimated earthworm effect in categories with a non-significant effect (i.e., cells with sandy soils, legumes, or receiving $> 30 \text{ kg N ha}^{-1} \text{ yr}^{-1}$).

- Line 142 "We then selected the most important drivers of the earthworm effect" – it is not clear how this was decided/done. The underlying meta-analysis provides many other coefficients that could have been used in this synthesis analysis (e.g., changes in climate, which are often found to be important). This should be expanded upon in terms of why those predictors were chosen and other not, and what implications this may have.

RESPONSE: We now provide additional detail about how we selected the variables included in the analysis. In the Methods section we explain that we "selected the most important drivers of the earthworm effect by examining the effect sizes reported by van Groenigen et al.⁸ and considering plant and environmental factors where the categories showed clearly differing earthworm effects (e.g., no-effect vs. positive effect, as reported for high vs. low N application) or had non-overlapping confidence intervals for at least two of the categories (e.g., $\text{pH} > 7$ vs. $\text{pH} \leq 7$). These variables were then cross-referenced against available global data layers, such that we ignored potential drivers for which there is no available data set (e.g., crop residue application rate). The factors thus selected included: crop type, soil pH, soil texture, nitrogen (N) application rate, and earthworm abundance (individuals m^{-2})."

We note factors such as soil organic matter content and climate were excluded by our approach, since there were not large differences between the different levels (i.e., confidence intervals were overlapping; see van Groenigen et al. (2014) Supplementary Table 3).

- Finally, how was the non-linear function for earthworm abundance calculated? In the underlying meta-analysis, there are three coefficients for earthworm abundance, so how robust is a function that has been created from only three data points?

RESPONSE: This is a good point, however, we note that there are actually four levels of earthworm abundance reported by van Groenigen et al. (2014): < 100 , $100-200$, $200-400$, > 400 ind. m^{-2} . We understand the concern here, as we tried several different approaches before we settled on using a continuous non-linear function. We originally generated coefficients using the same approach as for the other factors, but given that the vast majority of the cells in the earthworm density data layer contained < 100 ind. m^{-2} , and the fact that the observed effect was unexpectedly high for this category, we felt that this might have exaggerated the earthworm effect. In choosing a continuous function to fit the four points we went with a power function (see Table 1) that best addressed the higher than expected effect size for the < 100 ind. m^{-2} category. We now feel that this is fairly-well explained in the Methods section: "Earthworm

abundance, a , was calculated as a continuous, non-linear function that provided a best-fit for the available data (Table 1). We use this approach for earthworms rather than categories of abundance (as reported in the meta-analysis) since earthworm abundance for most cells in the global earthworm map was below the threshold for the 'low' earthworm density category (100 individuals m^{-2}) reported by van Groenigen et al.⁸. We consider our approach somewhat conservative since it allows cells with just a few individuals per m^2 to have a near zero effect of earthworms."

We are not sure what additional detail we could add to address the reviewer's concern here but are happy to provide more information if needed.

- As far as I can tell, the underlying meta-analysis tested each subgroup with a single model (e.g., one model on the effect of soil pH on yield, and another model on the effect of earthworm abundance on yield, etc. etc.). Fonte et al., have assumed that these coefficients (crop type, soil pH, soil texture, and crop-specific fertilizer N rates) can be multiplied together (i.e., a multiplicative effect, line 147). However, one could equally assume that these effects do not significantly interact with each other, and instead could also be causing an additive effect. Unfortunately, the van Groenigen meta-analysis can not be used to determine whether the equation should be multiplicative or additive, so I would suggest that the possibility is tested, and see how much the results change as a result of this assumption.

RESPONSE: We agree that there are several key limitations with our approach and with nature of the underlying data provided by van Groenigen et al. (2014). We also feel that the reviewer touches on an important point here, but we are not sure that we completely understand what they mean by 'multiplicative' vs. 'additive' effects. We note that by simply multiplying the coefficients together, our approach assumes that there are no interactive effects between factors – i.e., the effect of one factor does not depend on the level of another factor. It should be noted that when embarking on this study we contacted van Groenigen et al. and obtained the original data set used for the meta-analysis, with the hopes of better elucidating some of the potential interactions between factors, but after further exploration we concluded that there were simply not enough datapoints to tease apart interactive effects and we opted for the approach used here.

To address the Reviewer's concern, we have not added the following sentence to the Results and Discussion that touches upon this limitation: "Additionally, we point out that our analysis assumed simple additive effects for the plant and environmental factors that influenced the earthworm benefit, as we were not able to parse out potential interactions between these drivers."

Minor point:

Line 76-77: "...due *to* the vastly..."

RESPONSE: correction made

Reviewer #2 (Remarks to the Author):

This study is interesting as it shows clearly that earthworms contribute to food production at the global scale. They showed with a meta-analysis that in the whole world the Impact of earthworm on global agriculture is 6.5 % for global grain 2.3 % of legume yield.

It is true that up to now, there is not serious measurements of the earthworm potential contribution to global agricultural production. Most of the information comes from small trials and we don't have the idea of their impact at large scale.

The results of the analysis are interesting, we see that they impact cereals and legumes differently. As well their impact is more important in the global south although they have impact globally. They suggest that if there is an improved management of soil biological communities, the impact of earthworm will be larger. I Like that they mention that it can be some biases in their analysis which can give some over or under estimations.

The methods used are well explained although I am not an expert on this kind of analysis.

The manuscript is well written and clear. I think this communication is important as it is one of the first work that gives the idea of the impact of earthworms and soil biodiversity can have in food production at a global scale, which is a proof of the importance of agro -and soil biodiversity for decision makers.

So, I recommend the publication of this communication without any corrections,

I hope that over time more data will be available to draw more robust and secure conclusions as their data comes from 54 studies (462) points.

Kind regards

RESPONSE: We appreciate the reviewer's kind words and encouragement.

Reviewers' Comments:

Reviewer #1:

Remarks to the Author:

Thanks to Fonte et al for making the changes to their manuscript as well as clarifying points within the cover letter.

I am still happy with the manuscript, so I am just responding to a couple of previous points I raised.

- For the edits to the 'non-linear function for earthworm abundance', I would just add into the main text that it is a power function. This information can provide a lot of insight into the form of the shape, which will highlight the remaining of the information given about the non-linear relationship.

- I know see the issue for the additive vs. multiplicative effects! The main text now reflects your code, as from looking again I can see you did use an additive model. However, unless I am mistaken, the equation you give in the main text, is indicating a multiplicative model (a '*' between the predictor terms). A simple edit to change the '*' to a '+' and I would very much agree with everything you have written.

Again, it's a cool paper, and I am glad an analysis like this has been done!

Responses to Referee Comments

Reviewer #1 (Remarks to the Author):

Thanks to Fonte et al for making the changes to their manuscript as well as clarifying points within the cover letter.

I am still happy with the manuscript, so I am just responding to a couple of previous points I raised.

- For the edits to the 'non-linear function for earthworm abundance', I would just add into the main text that it is a power function. This information can provide a lot of insight into the form of the shape, which will highlight the remaining of the information given about the non-linear relationship.

RESPONSE: We now clarify that we used a "continuous, non-linear (power) function" for the coefficient related to earthworm abundance.

- I know see the issue for the additive vs. multiplicative effects! The main text now reflects your code, as from looking again I can see you did use an additive model. However, unless I am mistaken, the equation you give in the main text, is indicating a multiplicative model (a '*' between the predictor terms). A simple edit to change the '*' to a '+' and I would very much agree with everything you have written.

RESPONSE: We remain unclear about the reviewer's concern here. If we are not mistaken, our model is indeed additive because it does not consider interactive effects. For example, the effect of fertilizer application does not depend on soil texture, but rather operates independently of the other factors. Regardless, we feel that our coefficients should be multiplied together. Based on the way we have constructed them, each coefficient acts as a scalar to either increase or decrease the earthworm effect based on the level of each factor considered.

To clarify the multiplication steps in the code, we have added the comment, "All coefficient layers are multiplied to investigate total earthworm effects" to line 92. Within each respective crop group block, this function is written in lines 132-133 (grains) and lines 185-186 (legumes). The later summing steps (i.e., lines 146-151 in grains and lines 199-204 in legumes) are calculating absolute change in crop production values. These changes can be found in the same Dropbox files (<https://www.dropbox.com/sh/icbv07bhm7dig7m/AAA2xCmFM5K3OqvBQ4PcVxHma?dl=0>) and will also be publicly available in our Zenodo repository (10.5281/zenodo.8235224).

Again, it's a cool paper, and I am glad an analysis like this has been done!

RESPONSE: We appreciate the supportive comments.